# Weeds: An Insidious Enemy or a Tool to Boost Mycorrhization in Cropping Systems?

**DOI:** 10.3390/microorganisms11020334

**Published:** 2023-01-29

**Authors:** Alessandra Trinchera, Dylan Warren Raffa

**Affiliations:** Council for Agricultural Research and Economics (CREA), Research Centre for Agriculture and Environment, Via della Navicella, 2, 00184 Rome, Italy

**Keywords:** mycorrhizal fungi, spontaneous flora, service crop, mycelial network, agroecological practices

## Abstract

Weeds have always been considered an insidious enemy, capable of reducing crop production. Conversely, the agroecological vision attributes a key role to the spontaneous flora in promoting plant diversity and belowground interactions, which may improve the ecological performance of agroecosystems. We summarized the literature on the weeds’ arbuscular–mycorrhizae (AM) interaction and we analyzed evidence on the: (i) AM suppressive/selective effect on weed communities; (ii) effect of weeds on AM colonization, and (iii) positive role of AM-supporting weeds on forming shared mycorrhizal hyphal connections in agroecosystems. While some authors conceptualized AM as a weed control tool, others underlined their selective effect on weed communities. Recent studies suggest that AM-host weeds can participate in the development of a common mycorrhizal mycelial network (MMN) among different plants species. Nevertheless, direct evidence of the actual exchange of nutrients and C between coexisting plants through MMN in agroecosystems is missing. Although the effect of agricultural practices on plant community-AM interactions are complex, more conservative farming management seems to foster AM populations. Future studies should focus on: (i) field studies, (ii) weed communities and their traits, rather than on the most abundant species, and (iii) the use of advanced analytical techniques, able to monitor MMN development and functionality.

## 1. Introduction

When asking farmers how they usually manage weeds, the most frequent approach is to eliminate or strongly contain them. The capacity of spontaneous flora to rapidly colonize a fertilized cropping system induces hard competition with the main crop for available water and nutrients, sometimes in favor of the weeds [1]. The interference between the main crop and weeds is the result of the competition for water and soil nutrients, and the allelopathy among the different plant species coexisting in a given agroecosystem [2]. Starting from 1990, several studies approached such a complex plant relationship [3,4,5]. More recently, some researchers focused on the capacity of mycorrhizal fungi, and particularly of arbuscular mycorrhizae (AM), to strongly influence the community composition of spontaneous flora in cropping systems. AM can affect the nature of weed communities in agroecosystems in different ways, including changing the relative abundance of mycotrophic AM-host weed species and non-mycotrophic non-host species [6,7]. When competing, weeds and crops modulate their behaviors in relation to soil microorganisms, showing a greater dependence on associations with the soil symbiotic microbiota to increase their growth [8].

The current agroecological approach attributes an important role to AM-host weeds, or spontaneous flora, in designing sustainable and performing agroecosystems [9,10,11]. Simard et al. (2012) [12] found that different forest species are able to develop a belowground common mycorrhizal mycelial network (MMN), which implies also material transfer, such as organic C, nutrients, water, defense signals, and allelochemicals among AM plants. From this finding, we can hypothesize that similar mechanisms could also occur in herbaceous systems, where mycorrhizal flora can contribute to the development of a MMN. Our hypothesis is that, in presence of AM fungi, the infection of mycorrhizal weeds could facilitate the connections among external AM hyphae by creating a shared mycelial network, thus boosting the mycorrhization in the whole agroecosystem, also benefiting the main crop. However, despite the growing research interest on the agroecological role of weeds in mediating AM colonization and, hence, contributing to improve agroecosystem functioning, a comprehensive analysis of the literature on weeds–AM fungi relationships is still missing.

In this study, we reviewed the literature on weeds and AM symbiosis to analyze evidence on: (i) AM as a weed control practice and/or as a selector of specific weeds assemblages; (ii) the effect played by weeds on AM infection in cropping systems, and; (iii) the role of supporting arbuscular mycorrhizal (SAM) weeds in fostering AM hyphal connection in agroecosystems.

## 2. Methodological Approach

The systematic literature review was conducted on mycorrhizal fungi in relation to weeds using the online Scopus–Elsevier database (http://www.scopus.com, accessed on 1 January 2023). The query was carried out in October 2022 and included the following terms: “mycorrhizal fungi” OR “mycorrhizal” OR “mycorrhiza” OR “hyphal network” OR “AMF” OR “mycorrhizal network” OR “mycorrhization” AND “weed” * OR “spontaneous flora” in titles, abstracts, and keywords. All the studies related to forestry were excluded, except for papers dealing with agroforestry. Similarly, studies that dealt exclusively with mycorrhiza or weeds and not with their interactions were also excluded.

## 3. Results

The search yielded a total of 475 papers. The first article on the topic was published in 1975. Since then, the literature grew quite slowly, with only 18 studies published from 1975 to 1990 (Figure 1). An increasing interest in the weeds–mycorrhiza interaction was observed from 1991 to 2002, when 47 papers were found. The early 2000s saw a relevant increase in the number of papers on the topic (163 papers published from 2003 to 2012), but it was from 2013 to 2022 that the scientific interest in the weeds–mycorrhiza interaction rocketed with 242 studies published.

Europe showed the highest interest in the research on weeds–mycorrhiza interaction. Forty percent of the papers found came from this continent, with England being the most productive country. Papers from North America and Asia accounted for 26% of the total papers produced on the topic. The US was the most prolific country both globally and in North America, with a total of 120 published papers. China accounted for 45 studies published on weeds–mycorrhiza interaction and was the most important publisher in Asia. The rest of the papers were published by scientists from Oceania (6%), Africa (5%), the Middle-East (2%), and Latin America (1%) (Figure 2).

It should be noted that a vast number of tests were carried out in confined systems (microcosms, pots), and less frequently in open fields. Moreover, publications often focused on the effect of mycorrhiza on one selected weed species, generally the most abundant and impacting one, and not on the whole weed community. Finally, in most of the analyzed studies, the plant mycorrhization was induced via soil AM bioinoculants, neglecting the role of the native mycorrhizal fungi groups.

## 4. Discussion

We examined the results extrapolated from the literature to comprehensively address the issue concerning the relationships between mycorrhizal fungi, weeds, and their interactions in agroecosystems, and in particular: (i) the effect of mycorrhizal fungi on controlling/selecting spontaneous flora in cropping systems; (ii) the effect of weed community on AM infection in cropping systems, and; (iii) the development of a shared MMN among coexisting plants in cropping systems.

### 4.1. Do AM Control Weeds or Select Beneficial Flora?

#### 4.1.1. AM to Control Antagonist Weeds

One of the approaches applied when investigating the relationship between AM fungi and weeds is based on the application of mycorrhizal infection as a tool to control weeds. One of the first pieces of evidence of the capacity of AM fungi to contain spontaneous flora was observed in turfgrasses of the *Agrostis stolonifera* L. species. When AM fungi were abundant in the field, *Poa annua* L. was rare while *A. stolonifera* increased. These results could be due to a decline in AM fungi population due to *P. annua*, or to the AM fungi that could have instead altered the competition between the two grasses in favor of the mycorrhizal *A. stolonifera*, or limited the growth of *P. annua*, due to the fungi antagonistic effect [13]. This can potentially counteract the crop yield loss to weeds, limit the shift toward certain weed species’ predominance, and increase soil-beneficial organisms through a set of concomitant mechanisms [6].

The AM interaction was also reported to decrease weed seed germination. For example, a microcosm experiment showed that the seed germination of the *Orobanche* and *Phelipanche* species was reduced in the presence of root exudates from pea plants colonized by *Glomus mosseae* and *G. intraradices* [14]. In another microcosm experiment on sunflower, the total weed biomass was 47% less after the AM fungi inoculation of the main crop [15]. However, a previous pot experiment on striga (*Striga hermonthica* (Del.) Benth) and sorghum reported a decrease in striga emergence and an increase in sorghum production, following *G. mosseae* inoculation: specifically, striga emergence was reduced by 62% and the sorghum dry matter yield increased by 30% [16]. Another study obtained similar results on the AM effect on striga in sorghum and maize cropping systems [17]. The paper reported a significant amount of reduction in striga shoots, by 30%, and a reduction of more than 50% in maize and sorghum, respectively. Nevertheless, such a decrease in striga did not translate into higher cereal yields.

Overall, the effect of AM fungal taxa on weed growth seems to be not easily predictable. Concerning maize production, Li et al. (2019) [18] found that the effect of AM inoculants on weeds and maize growth varied greatly across plant species. Numerous AM taxa were shown to have negative effects on certain weeds but not on maize. These results suggest that the use of specific inoculants should be targeted to distinct species as a means of taking advantage of the selective effect of AM on weed communities.

The interaction between AM and weeds was also reported from experiments in a glasshouse. Vatovec et al. (2005) [19] studied the response of seedlings to AM inoculants of 14 agronomic weed species, sampled from three cropping systems (organic, transitional-organic, and high-input/conventional). The weed biomass response to AM fungi was highly variable across species, depending on specific weed traits, namely strong/weak-host and non-host species. The weed biomass response to inoculants was also significantly dependent on cropping systems, suggesting that highly conservative management, such as organic farming, can make the difference on AM fungi—weed interactions.

Two types of weed suppressive action were identified in different cropping systems: (i) direct effect, where weak host weeds were reduced as AM favored strong host weeds, and; (ii) indirect effect, where AM increased the competitive ability of strong host crops [20,21,22,23]. Based on these assumptions, the excellent meta-analysis by Li et al., (2016) [21] provided an array of management decisions based on the AM affinity of the crop and weeds, graphically represented in Figure 3.

#### 4.1.2. AM Selects Host Weed Assemblages

While most of the studies cited in the previous section reported on the use of AM inoculant for weed control, other papers highlighted the effect of AM to gradually select “useful” weeds or, in other words, weeds with AM-supporting traits. Cameron (2010) [24] stressed the ability of AM fungi in facilitating the shift towards mycorrhizal host plant communities and suppressing the non-mycorrhizal ones. This can benefit the whole agroecosystem and is particularly relevant when mycorrhizal crop species are grown.

The positive effect of AM fungi on selecting spontaneous flora was also found in grasslands. In these systems, Koziol and Bever (2017) [25] demonstrated that AM inoculants could promote plant diversity, and select functional weed assemblages able to provide a restoration outcome, while inhibiting less desirable weedy plants.

In agroforestry, the contribution of plant diversification to weed management appeared to be mediated by AM fungi; integrating a *Faidherbia albida* tree into a sorghum cropping system maintained the persistence of AM fungi that protected the main crop from *Striga hermonthica* colonization [26]. One possible explanation of these selective mechanisms played by AM fungi relates to their ability to modulate phosphorus accessibility to plant roots. It was observed that, when the level of plant-available phosphorus is low, AM fungi colonization favors the P uptake by mycorrhizal species. As a result, the non-host species’ development is hindered by the combined effect of the low P availability and the high competition with AM-host species which can severely impact the non-host seedbanks in the long run [27,28]. Although the plant host trait has been indicated as a main criterion for the selective effect of AM on weed assemblages, the study by Säle et al., 2022 [29] reported that plant host trait does not always guarantee a positive contribution to the agroecosystem mycorrhization. Based on different AM fungi isolated from some European cropping systems, it was found to have a negative or neutral effect on aboveground biomass of *Echinochloa crus-galli*, *Solanum nigrum* and *Papaver rhoeas*, regardless of their AM-host/non-host trait. The authors concluded that some weed species do not benefit from AM fungi in terms of growth regardless of their traits, thereby strengthening the AM role in weed selection.

Overall, the literature analyzed suggested that scientists conceptualized AM in the context of weed management in two primary ways: (i) as a tool to control weeds or (ii) as a biological tool to select specific weeds. While the former originated from a conventional approach to weed control, the latter could offer a more comprehensive view of weed management that recognizes the ecological dynamics of AM–weeds interaction. This could be further operationalized at the field level and be considered as a starting point to select weeds able to support ecosystem services without affecting, or even improving, crop production.

### 4.2. Effect of Weed Community on AM Colonization in Cropping Systems

The other side of the coin concerns the role played by the whole weed community on mycorrhizal populations and the colonization of plants within cropping systems.

The fact that several weeds are AM hosts, as reported by numerous studies, suggests that weeds hold a great potential to foster AM in agroecosystems. Among those, Ishii et al. (1998) [30] represents one of the first study focused on the whole weed community, where the presence and relative abundance of mycorrhizal host and non-host species were evaluated in terms of “cropping system mycorrhization”. In a citrus orchard, both spring and summer weeds as *Rumex acetosa* L., *Agropyron tsukuhiense* (Honda) *Ohwi var. transiens* (Hack.) Ohwi, *Stellaria media* Villars., *Vicia cracca* L., *Lamium amplexicaule* L., *Medicago polymorpha* L., *Erigeron canadensis* L., *Amaranthus lividus* L., *Cynodon dactylon* (L.) Pers., *Beckmannia syzigachne* (Steud.) Fern., *Commelina communis* L., *Oxalis corniculata* L., and *Digitaria adscendens* Henr.) were highly colonized by mycorrhizal fungi: particularly, in spring season, AM spores ewere mostly found on roots of some mycorrhizal host weeds, such as in *Stellaria media*, *Vicia cracca*, and *Lamium amplexicaule*. On the other hand, no AM infection was observed in *Rumex japonicus* Houtt., *Equisetum arvensis* L., and *Polygonum blumei* Meisn, as these species are usually recognized as non-mycorrhizal weeds.

Given the effect of weed communities on AM population, weed control practices can also negatively affect the AM population in agroecosystems. In coffee production, weed community management was one of the key factors influencing the composition and abundance of AM spores [31]. Similarly, soil solarization was reported to significantly reduce AM weed hosts, and hence AM, with possible negative effects on AM crop colonization [32].

In general terms, highly diversified systems showed a significantly higher crop mycorrhizal colonization, compared to monocropping. Some examples concern (i) oil palms, by comparing monoculture to agroforestry [33]; (ii) different herbaceous crops [*Sorghum bicolor* L. Moench (sorghum), *Ambrosia artemisiifolia* (L.) (ragweed), and *Amaranthus powellii* S. Wats. (pigweed)], grown as monocultures or in mixtures [34], (2021), and; (iii) maize [18], where a weeded system was compared to an non-weeded one, where *C. album*, *E. crus-galli*, and *V. arvensis* were the most abundant species [35].

A huge effort has also been put into collecting data concerning weeds and crop traits related to AM (e.g., [36]). Those studies are of fundamental importance to study the functionality of weed assemblages related to AM and their potential to act as “AM providers” to crops. Although those interactions are complex and variable, evidence of the positive effects of AM-host weeds on crop mycorrhization was reported. The first step towards the idea of a common fungal network among herbaceous plants and/or crops appeared when the extra-radical mycelium was observed on AM-host weed roots in arable cropping systems. Such interaction was reported to be critical to boost AM colonization of young wheat seedlings, both in the greenhouse and in organic cropping systems [10,37]. The use of diversification/conservative agricultural practices was reported to further increase weed communities able to support AM and contribute to the development of a common mycelial network with a positive effect on crops. Clear supporting evidence is provided by Ramos-Zapata et al. (2012) [38], who carried out a long-term experiment on maize production in the tropics. The authors found that the use of mycorrhizal cover crops (*Mucuna deeringian* and *Lysiloma latisiliquum*) had a positive impact on mycorrhizal weed species’ richness as well as on the percentage of plant roots colonized by AM fungi in field, increasing maize production in the long run [38]. These results are in line with those obtained by Feldmann and Boyle (1999) [39], who studied the role of host or non-host mycorrhizal traits of weeds in maize monocropping systems, both in field and greenhouse conditions. Specifically, the absence of any accompanying flora led to a loss of mycorrhizal spore types in the AM fungi community, lower effectiveness of the persistent AM fungi populations, and a decline in maize biomass production.

Similar results were obtained for horticultural crops and orchards. AM-host weeds had a key role in improving lettuce mycorrhization with possible positive effects on crop production and quality [40]. Likewise, AM spores grew significantly with the increase in the number of weed species in subtropical citrus and Mediterranean olive orchards, thereby contributing to the mycorrhizal colonization of crops [31,41,42].

Nevertheless, the net positive effects of the higher AM biomass triggered by communities of AM-supporting weeds on the main crop were not always confirmed. For instance, the introduction of different cover crops in vineyards increased mycorrhizal colonization and AM fungal spore populations on spontaneous flora but did not positively affect mycorrhization of grapevine roots, probably due to the missed contact between colonized roots and excessive disturbance by tillage [43,44].

Finally, these results suggest that weed community composition and diversity, the cropping system, and the management practices are complementary key drivers, acting by supporting or inhibiting mycorrhization. The interactions across AM weeds and farming practices are complex and variable across production systems. Still, a growing body of literature is highlighting how weeds, if well managed, could be considered “service crops” able to foster AM colonization across the elements of the agroecosystem and potentially determine the final performance of the main crops.

### 4.3. The Agroecological Approach Applied to Cropping Systems: Extra-Radical Hyphal Connection and Hypothesis of Mycorrhizal Mycelial Network (MMN) Development in Herbaceous Systems

The first studies that explored the AM fungi infection of different plant species date from 1990, when several pot experiments (*Chenopodium album* L., spring wheat, lettuce) and field trials (*C. album*, *Galinsoga parviflora* Cav., *Sinapis arvensis* L., *Sonchus oleraceus* L., spring wheat, and maize) showed that the AM colonization of weeds was increased in the presence of mycotrophic crops, taking place only when growing in the vicinity of the main crops [45]. On the contrary, weed AM infection decreased in the presence of facultative mycotrophic crops. Here, the first indirect evidence of the importance of the “proximity” between the root systems of SAM species to boost the formation of a shared fungal mycelium was observed. Nevertheless, this mechanism was not yet well explored in open-field, and particularly on herbaceous systems. Under the agroecological vision, the cropping system is constituted by a community of different plant species, as a result of a “suite” of functional traits, where the main crop, weeds, and AM fungi work together for optimizing the use of available water and nutrient resources [46,47]. Among those functional traits, the SAM trait drives the plant selection in the field in favor of those species able to mycorrhize, assuring them of an increased P uptake and an improved yield and quality [11,48,49].

Although the role of a mycorrhizal mycelial network (MMN) on biogeochemical cycling, plant community composition, and ecosystem functioning was deeply studied in natural systems [9,12,50,51], it has not been widely investigated in agroecosystems. Recently, in a greenhouse potted experiment, the effects of AM association on the interference of *Bidens pilosa*, *Urochloa decumbens*, and *Eleusine indica* on soybeans was evaluated under plant competition, with or without contact with roots of another species. Positive interactions between soybean mycorrhizal colonization and competing host plants were found, irrespective of weed species or root contact, although direct evidence of the exchange of C and nutrients across plants were not provided [52]. Apparently, in a confined pot system, AM fungi colonize all plants, independently from the plant species and their traits. Evidence of the aspecific AM root colonization capacity in the field, especially when tillage is reduced, was possibly found by Oehl and Koch (2018) [53]. The absence of tillage positively affected AM fungi diversity in different Chinese vineyards compared to tilled systems, regardless of the inter-row *Lolium perenne* cover, the level and type of P fertilization, apparently contrasting with the results obtained by De Cauwer et al. (2021) [27]. In parallel, on a multi-year alfalfa-winter cereal rotation, the highest mycorrhizae abundance and diversity were associated with a continuous ground cover of herbaceous plants and infrequent tillage, compared to frequently tilled plots [54]. The decrease in AM-seedbanks’ diversity in disturbed soils confirmed that tillage can favor i) ruderal and disturbance-tolerant AM fungi taxa and ii) a selection of SAM species able to be colonized only by these taxa [55]. However, the presence of SAM weeds does not always guarantee the increase in mycorrhizal infection in the agroecosystem, as the soil mechanical disturbance is a limiting factor, due to the interruption of hyphal connection among coexisting plant roots [56].

Several theories have been developed to explain the advantages for SAM plants which could form a common extra-radical hyphal connection. A first theory assumes that SAM plants can promote the mycorrhizal infection of neighboring seedlings, thereby acting as an inoculum to favor C supply and maintain a common MMN among plants [57]. A second theory focuses on the role of MMN in equally distributing nutrient resources via plant interconnection in a given ecosystem [58]. Unfortunately, measuring the transfer of nutrients, energy, or signals from one plant to another mediated by MMN in field is really challenging, and currently there is no reference method to assess the MMN functionality and development. Nevertheless, the extra-radical mycorrhizal mycelium formed by coexisting plant species—which does not always imply the internal AM infection of involved plant roots—can be observed using electron scanning microscopy. A first attempt to indirectly quantify the common extra-radical mycorrhizal mycelium was applied in Mediterranean organic winter cereal cropping systems, where the mycorrhiza-mediated interference between crops (rye or spelt) and weeds was studied. The contribution made by crops and specific weed species to promote mycorrhization led to the formulation of the quantitative indicator “mycorrhizal colonization intensity of the agroecosystem” [59].

Another evidence of the common extra-radical mycorrhizal mycelium in diversified systems was found when investigating non-SAM species. It is well known that the ability of *Brassicaceae* and *Chenopodiaceae* to exudate specific allelochemicals from roots inhibits the branching of AM fungi hyphae onto root external cells [60]. However, it was found that mycorrhizal extra-radical mycelium can cover also roots of weed species belonging to the *Brassicaceae* family: on these roots, morphological types of arbuscular and coiled hyphae were observed, which are the most common in grasses [61]. Similar results were obtained in an organic beetroot intercropping system: here, an extra-radical mycelium was found on beetroot roots surfaces as a result of the presence of SAM weeds (namely, *Capsella bursa-pastoris* L., *Senecio vulgaris* L., *Spergula arvensis* L., and *Plantago maior* L.) in intercropping, while the external hyphal mycelium was missed in the monocropping system, where SAM species were not recorded [62].

The common extra-radical mycorrhizal mycelium was also observed in an organic *Cocumis melo* L. production system, mulched with flattened durum wheat [11]. Here, in *Polygonum aviculare* L., *Anagallis arvensis* L., *Rumex crispus* L., *Convolvulus arvensis* L., and, secondarily, *Plantago media* L. and *Sonchus oleraceus* L., relative abundances were strongly correlated to the melon mycorrhizal colonization and to the common extra-radical mycorrhizal mycelium development, when compared to not mulched or weeded systems. The shared extra-radical mycelium was observed on undisturbed melon root fragments collected at melon harvesting, using Scanning Electron Microscopy (SEM) under variable pressure, equipped with a LaB_6_ electron sources and back-scattered electrons detector (Figure 4).

This visual indirect evidence of a hyphal connection among different plant individuals allows us to hypothesize that a functional MMN could be potentially developed not only among different tree species, where ectomycorrhizal fungi are prevalent in colonizing roots [12,50], but also in herbaceous systems, where AM fungi dominate the plant-fungal symbiosis.

In systems where AM-host cover crops were introduced (e.g., rye or spelt), the presence of a spontaneous mixed and highly diverse flora addresses the mycorrhizal colonization towards SAM weeds. Our interpretation attributes to SAM weeds the role of “functional internodes” within a common extra-radical mycorrhizal mycelium, thus promoting the mycorrhization and the growth of the following crop [11,63]. Conversely, when the non-SAM weed species dominate, the extra-radical mycelium develops only by covering the surface of non-SAM roots with coiled hyphae [61] and the surface fungal mycelium. In this case, the non-SAM weeds behave as “unfunctional internodes” within the common extra-radical mycorrhizal mycelium, guaranteeing a certain spatial AM hyphal continuity (Figure 5).

In cropping systems dominated by the spelt cover crop, *Rumex crispus* L., *Stellaria media* L., *Veronica persica* L., *Polygonum aviculare* L., and *Anagallis arvensis* L. SAM weeds were the most abundant. Conversely, in a non-weeded system, the non-SAM weeds’ abundance increased, thereby decreasing the mycorrhization of the whole agroecosystem [59] (Figure 5).

Those findings suggest that weeds are much more than a set of spontaneous species in constant competition with crops. Rather, weeds can strongly support the functionality of the agroecosystem, depending on their specific traits. Specifically, SAM species can both promote the growth of belowground fungal mycelium, useful to colonize crops via the AM extra-hyphae expansion or favor the AM mycelial connection among individual crops. Evidently, weeding practices can strongly impact native seedbanks’ diversity, potentially affecting the abundance of SAM weeds in the field and, consequently, limiting the fungal mycelium development and the ecosystem services provided by mycorrhizae. On the contrary, the introduction of SAM cover crops, such as the winter cereals [59], or the intercropping with legumes (e.g., faba bean, [62]) can promote the AM colonization of SAM weeds and, thus, of the whole cropping system.

## 5. Conclusions

This study analyzed the scientific literature on weeds and AM colonization. Firstly, two main approaches were followed to study the weed–AM interaction. While some scientists conceptualized AM as a tool to control weeds, others underlined the selective effect of AM on weed communities.

Secondly, the literature indicated that weeds have a critical role in mediating the population of AM in agroecosystems, thereby suggesting that weeds could be considered as “service crops” rather than solely a worrying element in agricultural fields.

Thirdly, the AM ability to colonize the weed community in agroecosystems is a result of complex interactions belowground among plants and mycorrhizal fungi, where weed diversity, and particularly SAM traits, drive the success of the mycorrhizal infection. Although we found strong evidence of a common extra radical mycelium across different crop and weed species, studies that unequivocally verified the C and nutrients exchange mediated by MMN in agroecosystems were not found.

Concerning agricultural management, farming practices act as an additional filter to weed communities and AM development. Mycorrhizal cover crops, intercropping crops with SAM traits, and the reduction/absence of soil tillage are agroecological practices that can promote the extra-radical hyphal connection and possibly the development of a MMN among coexisting plants. To this end, knowledge of weed community functional traits is instrumental to inform on (i) mechanisms underlying the development of a functional MMN in the field and (ii) other ecological services supplied by spontaneous flora as a means to increase the agroecosystem performances (e.g., productivity, support to pollinators, nutrient cycling).

Finally, we would stress the importance of future research addressing:−In-field studies more than in confined systems, the latter not always being representative of the actual interactions among neighboring roots of plant species;−Experiments mostly focused on the effect of management on the whole weed community and its traits, rather than on the most abundant species;−Advanced analytical techniques, able to monitor the development and functionality of MMN among roots at the belowground and the soil C supply and P availability fluctuation.

## Figures and Tables

**Figure 1 microorganisms-11-00334-f001:**
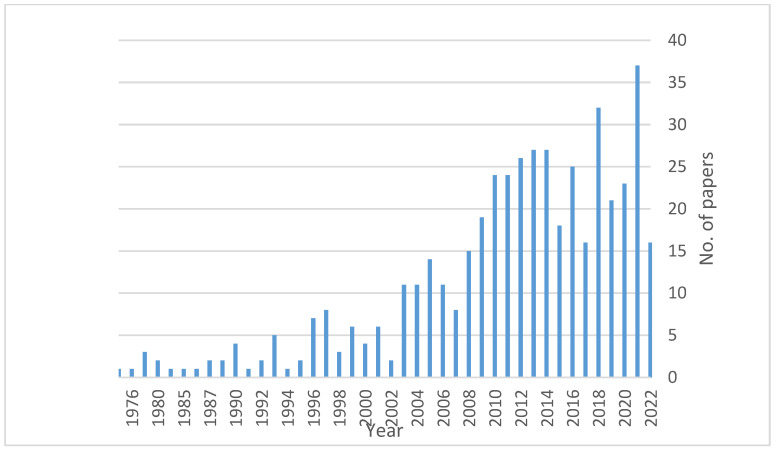
Annual distribution of the paper published on weeds–mycorrhiza interaction from 1975 to September 2022.

**Figure 2 microorganisms-11-00334-f002:**
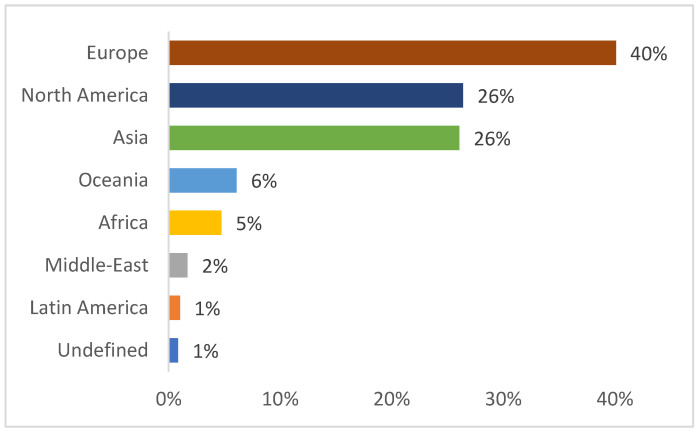
Share of geographical distribution of the paper published on the weeds–mycorrhiza interaction from 1975 to September 2022.

**Figure 3 microorganisms-11-00334-f003:**
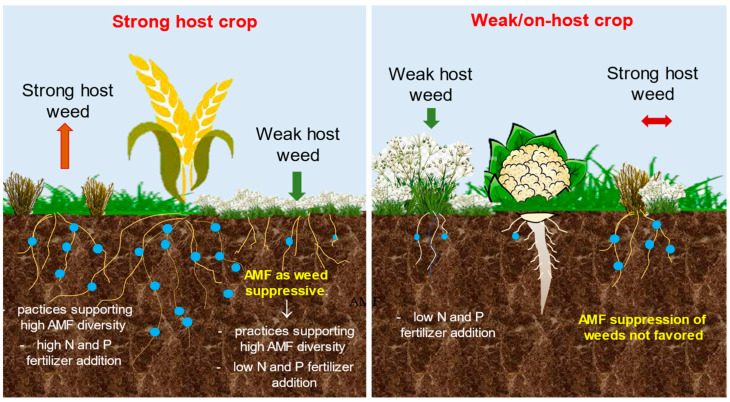
Recommended management decisions based on host status, and crop-weed community composition for use of arbuscular mycorrhizal fungi (AMF) in weed control. Adapted from [19].

**Figure 4 microorganisms-11-00334-f004:**
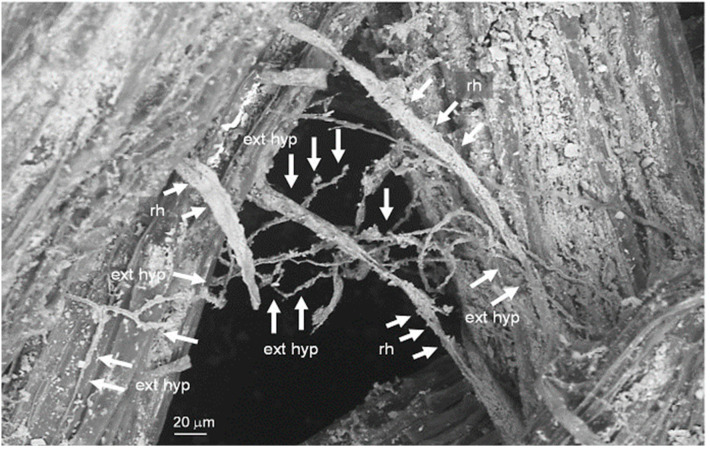
SEM image of extra-radical hyphae forming a shared mycorrhizal mycelium on fine lateral roots of *Cucumis melo* L., grown on flattened spelt. ext-hyp: AMF extra-radical hyphae; rh: root hairs. Magnification = 1.0 KX.

**Figure 5 microorganisms-11-00334-f005:**
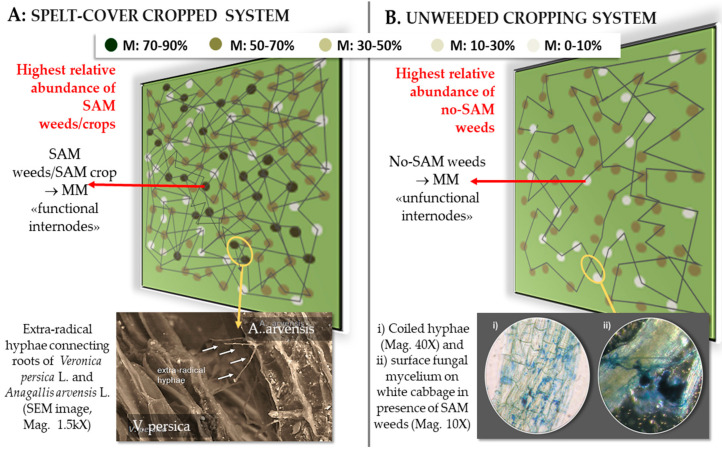
Comparison between common extra-radical mycorrhizal mycelium (MM) in two horticultural cropping systems differently managed (CREA-OF, Monsampolo del Tronto, AP, Italy): A. Weed control using spelt (*Triticum dicoccum* L.) as cover crop: electron scanning microscopy shows connecting hyphae between *Veronica persica* and *Anagallis arvensis*, both SAM weeds. B: non-weeded system, without cover crop (control): optical microscope images show coiled hyphae and surface, non-colonizing fungal mycelium on no-SAM *Amaranthus retroflexus*. Area = 1.0 m^2^. Total dots in each area correspond to the total number of plants×m^−2^ (∑ main crop + weeds/m^2^) counted in the field. Each dot represents a single individual plant (cover crop + weeds in A, weeds in B). Increasing color darkness corresponds to an increasing range of mycorrhizal colonization intensity of each plant species (M% ranges, as reported). Year: 2015. (Modified from: [58]).

## Data Availability

No new data were created or analyzed in this study. Data sharing is not applicable to this article.

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
