# Peer review of "Weeds: An Insidious Enemy or a Tool to Boost Mycorrhization in Cropping Systems?"

_microorganisms, 2023, doi:10.3390/microorganisms11020334_

Round 1

Reviewer 1 Report (Previous Reviewer 2)

I think this review has important implications for mycorrhizal and agricultural ecology and will be of interest to many researchers. Current manuscript has been revised appropriately and is much better than the first manuscript submitted.

Reviewer 2 Report (Previous Reviewer 3)

The revised manuscript can be published.

This manuscript is a resubmission of an earlier submission. The following is a list of the peer review reports and author responses from that submission.

Round 1

Reviewer 1 Report

Although the present topic is interesting, Table 1 and  Figure 3 were directly copied from the original references without any modification, which is unacceptable to me.

Author Response

Dear Reviewer 1,

thank you for your review. Kindly find below the reply to your comments.

Table 1 was replaced by an original figure (new Figure 3, made by authors), representing the concept expressed into the reference paper Li et al., 2016 (19).

Figure 3 was replaced with a new original figure (now, Figure 4),  supplied by the authors.

Regards

Alessandra Trinchera and Dylan Warren Raffa

Reviewer 2 Report

As the cost of fertilizers and fuel increases, attention is turning to methods of growing crops that rely on soil fertility. The least expensive, sustainable way to increase soil fertility is not the introduction of green manure, cover crops or compost, but the proper use (management) of weeds. The content of this review is very timely as it is required in the current social situation. The phenomenon of weeds supporting crop growth through the enhancement of AM networks is conceptually easy to understand, but there are few scientific descriptions of this phenomenon. It is still an unexplored research field to discuss it, and a new approach to research is needed altogether. It is highly appreciated that the authors mention these difficulties and the future prospects. Overall, this paper is highly complete, but there are some mentiones that are recognized as difficult to draw from previous findings.

Line 96, endogenous -> indiginenous?

Lines 109-112, The causal relationship is not clear enough to say that the AMF directly affected the growth of Poa annua.

Line 127, Is it appropriate to describe it as specie dependent? Wouldn't it be more appropriate to say "context dependent" or "not predictable"?

Line 133, I am a little confused as to what SPECIFIC refers to.

Line 166, Is there clear proof of a causal relationship in that AM reduce the non-host seed banks?
Galinsoga quadriradiata is probably Asteraceae, not a non-host.

Lines 167-168, The causal relationship on this mention is not clear for me.

Lines 356-361, I am not sure what the authors are trying to say with this statement.

Author Response

Dear Reviewer 2,

thank you for appreciating our review and for your accurate review. Kindly find below the reply to your comments.

Line 96, endogenous -> indiginenous? R: We changed “endogenous” into “native”.

Lines 109-112, The causal relationship is not clear enough to say that the AMF directly affected the growth of Poa annua. R: Thank you. We modified the sentences according to your suggestions. The sentences were revised to make clearer the mechanism of competition and antagonistic effect between the two species mediated by mycorrhizal fungi.

Line 127, Is it appropriate to describe it as specie dependent? Wouldn't it be more appropriate to say "context dependent" or "not predictable"? R: As suggested, we changed the sentence into “not easily predictable”.

Line 133, I am a little confused as to what SPECIFIC refers to. R: We agree with your comment and deleted the word “specific” in the text.

Line 166, Is there clear proof of a causal relationship in that AM reduce the non-host seed banks? Galinsoga quadriradiata is probably Asteraceae, not a non-host. R: Thank you we understood your concern and modified the sentence to improve the clarity. We also confirm that Galinsoga quadriradiata is an Asteraceae plant and we modified lines 165-170, accordingly.

Lines 167-168, The causal relationship on this mention is not clear for me. R: Those lines were revised.

Lines 356-361, I am not sure what the authors are trying to say with this statement. R: The sentence was modified to make the concept clearer.

Regards

Alessandra Trinchera and Dylan Warren Raffa

Reviewer 3 Report

This is an interesting and important review dealing with environmental and practical aspects  of weeds-main crops-mycorrhizal (AM fungi) interactions.

Author Response

Dear Reviewer 3,

we thank you for appreciating our review.

Regards

Alessandra Trinchera and Dylan Warren Raffa